# Coumarin-Containing Light-Responsive Carboxymethyl Chitosan Micelles as Nanocarriers for Controlled Release of Pesticide

**DOI:** 10.3390/polym12102268

**Published:** 2020-10-01

**Authors:** Song Feng, Junqin Wang, Lihua Zhang, Qin Chen, Wang Yue, Ni Ke, Haibo Xie

**Affiliations:** Department of Polymer Materials and Engineering, College of Materials and Metallurgy, Guizhou University, Guiyang 550025, China; fszxy159@163.com (S.F.); Wjq010405@163.com (J.W.); qchen6@gzu.edu.cn (Q.C.); wangy6186enoch@163.com (W.Y.); NIKO.KN@139.com (N.K.)

**Keywords:** carboxymethyl chitosan micelles, light-responsive, coumarin, 2,4-D, controlled release, pesticide

## Abstract

Currently, controlled release formulations (CRFs) of pesticides in response to biotic and/or abiotic stimuli have shown great potential for providing “on-demand” smart release of loaded active ingredients. In this study, amphiphilic biopolymers were prepared by introducing hydrophobic (7-diethylaminocoumarin-4-yl)methyl succinate (DEACMS) onto the main chain of hydrophilic carboxymethylchitosan (CMCS) via the formation of amide bonds which were able to self-assemble into spherical micelles in aqueous media and were utilized as light-responsive nanocarriers for the controlled release of pesticides. FTIR and NMR characterizations confirmed the successful synthesis of the CMCS-DEACMS conjugate. The critical micelle concentration (CMC) decreased with the increase in the substitution of DEACMS on CMCS, which ranged from 0.013 to 0.042 mg/mL. Upon irradiation under simulated sunlight, the hydrodynamic diameter, morphology, photophysical properties and photolysis were researched by means of dynamic light scattering (DLS), transmission electron microscopy (TEM), UV-vis absorption spectroscopy and fluorescence spectroscopy. Moreover, 2,4-dichlorophenoxyacetic acid (2,4-D) was used as a model pesticide and encapsulated into the CMCS-DEACMS micelles. In these micelle formulations, the release of 2,4-D was promoted upon simulated sunlight irradiation, during which the coumarin moieties were cleaved from the CMCS backbone, resulting in a shift of the hydrophilic–hydrophobic balance and destabilization of the micelles. Additionally, bioassay studies suggested that this 2,4-D contained which micelles showed good bioactivity on the target plant without harming the nontarget plant. Thereby, the light-responsive CMCS-DEACMS micelles bearing photocleavable coumarin moieties provide a smart delivery platform for agrochemicals.

## 1. Introduction

The extensive utilization of pesticides in controlling weeds, pests and fungi greatly contributes to ensuring global agricultural productivity [1,2,3]. Nonetheless, excessive and repetitive use of pesticides has led to a series of adverse effects on environmental quality, safety of agricultural products and human health [4]. Controlled delivery technology has proven its potential for sustainable use of pesticides in agriculture [5,6,7]. The advantages of controlled release formulations (CRFs) of pesticides over conventional formulations include (i) reducing losses induced by evaporation and leaching [8,9], (ii) prolonging effective lifetime [10], (iii) protecting the active ingredients against environmental degradation by photo, heat, humidity and microorganisms [11,12,13], (iv) minimizing residues in the environment, agricultural products and food chain [14] and (v) reducing toxicity for nontarget organisms including humans [15]. Therefore, much academic and industrial effort has been devoted to the improvement of CRFs of pesticides that ensure the agricultural yields through an environmentally friendly approach [3,16,17].

Stimuli-responsive CRFs provide advanced and smart controlled release of pesticides, where the CRFs present responses to small external stimuli, resulting in changes in their chemicalor physical properties favoring the “on-demand” release of loaded pesticides [18]. Until now, several kinds of stimuli responsive CRFs have been fabricated using abiotic or biotic stimuli, such as enzymes [13], redox [19], pH [20], temperature [21] and light [22]. Compared with other stimuli, light as an external trigger for pesticide release has received a lot of attention, since it is a clean and remote stimulus that can be spatially and temporally controlled by tuning the intensity, wavelength and site of irradiation [23,24]. Light-controlled CRFs have strong potential applications in agriculture, able to take advantage of the abundant natural sunlight radiation in open fields, and even for greenhouses with simulated sunlight or UV light equipment [18]. Several light-responsive groups have been developed for this purpose, including coumarin [14], azobenzene [1], 2-nitrobenzyl [22] and spiropyran [25]. Among these light triggers, the coumarin-based platform is notably attractive because of the efficient photorelease ability and strong fluorescence property [26,27,28]. For example, Atta et al. reported a UV-light-responsive coumarin polymer for controlled release of pesticide 2,4-dichlorophenoxyacetic acid (2,4-D) [14]. Another distinct advantage of coumarin derivatives is the tunability of absorption wavelength by molecular design (*λ*_max_ from 310 nm to 490 nm), thus providing broader compatibility with light wavelength [26,29]. Reports have shown that the electron donating diethylamino substituents at the 7-position allows photochemical activity in the visible region, further allowing cleavage of the coumarin moieties from the polyphosphazene backbone upon visible light irradiation [30]. Xu et al. also reported a propesticide with 7-diethylamino substituted coumarin as a photocage for the release of insecticidal ingredients upon blue light or sunlight [31]. However, the current coumarin-based small molecule propesticide has certain obvious drawbacks such as its fast release rate and the need for organic solvents (i.e., methanol). Polymeric encapsulation seems to be a better choice in terms of prolonged and regulated release profile, eco-friendly aqueous media and multiple forms (i.e., capsules, micelles, microspheres and hydrogels) [10,22,32,33,34,35].

On the other hand, compared with synthetic polymers, the application of naturally occurring biopolymers (i.e., chitosan, cellulose and starch) as pesticide nanocarriers has a brighter future because of the desirable biodegradability, biocompatibility and adjustable physicochemical properties [33]. Typically, chitosan is among the most valuable biopolymers for the delivery of pesticides owing to its inherent antibacterial and antifungal activities [15,36]. It can even stimulate the innate immune systems of plants and improve resistance against disease and insects [37]. Chitosan molecules with enormous hydroxyls and amino groups can be readily modified for encapsulation and controlled release of agrochemicals, including herbicides, insecticides, plant growth regulators and fertilizers [38,39].

Thus, the present study was carried out aiming to develop light-responsive CRFs based on chitosan micelles for the controlled release of pesticides. This amphiphilic chitosan derivative was prepared by covalently conjugating hydrophobic light-responsive (7-diethylaminocoumarin-4-yl)methyl succinate (DEACMS) onto the main chain of hydrophilic carboxymethyl chitosan (CMCS) via formation of an amide bond. The self-assembled light-responsive micelles were utilized as nanocarriers for the controlled release of 2,4-D, a phenoxyacetic acid herbicide. We chose 2,4-D as the model pesticide because of its global use but potential risks to human health, and it has been identified as a class 2B carcinogen by the International Agency for Research on Cancer (IARC) [40]. The morphology, hydrodynamic diameter, photophysical properties and light-controlled release behavior of CMCS-DEACMS micelles were investigated upon simulated sunlight irradiation. Finally, the bioactivity of 2,4-D-contained micelles against one dicot target plant cucumber (*Cucumis sativus L.*) and one monocot nontarget plant wheat (*Triticum aestivum L.*) was also evaluated.

## 2. Materials and Methods

### 2.1. Materials

Selenium (IV) dioxide (SeO_2_), 7-diethylamino-4-methylcoumarin, sodium borohydride (NaBH_4_), succinic anhydride, potassium carbonate (K_2_CO_3_), 4-dimethylaminopyridine (DMAP), 1-ethyl-3-(3-dimethylaminopropyl) carbodiimide(EDC), *N*-hydroxysuccinimide (NHS) and 2,4-D were obtained from Aladdin Reagent Co. (Shanghai, China). Carboxymethyl chitosan with a degree of substitution of 0.8 was purchased from Shanghai Yuanye Biological Co. (Shanghai, China). The degree of deacetylation (DD) was determined to be 0.96, according to a method described in the literature [41]. Dioxane, tetrahydrofuran (THF), ethanol (EtOH), HCl, dichloromethane(CH_2_Cl_2_), chloroform (CHCl_3_), dimethylsulfoxide (DMSO), petroleum ether and ethyl acetate were purchased from Beijing Chemical Reagent Co (Beijing, China). Dialysis bag (MWCO, 3.5 kDa) was obtained from Union Carbide Co. (Danbury, CT, USA). Nitrogen was provided by Guiyang Shenjian Gas Company (Guiyang, China). Deionized water was used throughout the work.

### 2.2. Synthesis of 7-Diethylamino-4-Hydroxymethylcoumarin (1)

The 7-diethylamino-4-hydroxymethylcoumarin (1) was synthesized in a similar procedure according to the literature [30,42]. SeO_2_ (4.44 g, 40 mmol) was added to a solution of 7-diethylamino-4-methylcoumarin (4.62 g, 20 mmol) in dioxane (100 mL). The reaction mixture was refluxed at 90 °C under vigorous stirring with a protective atmosphere of nitrogen for 24 h, and then the reaction mixture was filtered and the filtrate was concentrated under reduced pressure. The obtained dark-brown residue was dissolved in 100 mL of THF/EtOH mixture (1:1), and NaBH4 (1.52 g, 40 mmol) was added slowly. The solution was stirred at room temperature for 4 h and then the suspension was carefully neutralized with 1 M HCl. Solvents were almost totally removed under vacuum. The residue was diluted with CH2Cl2 and extracted three times with 0.5 M K2CO3. The organic extracts were concentrated and purified by column chromatography (petroleum ether/ethyl acetate = 2:1) to obtain compound (1) as a reddish brown solid with a yield of 56% M.P. (°C): 131.6. ^1^H NMR (400 MHz, DMSO-*d*_6_, ppm): δ7.43 (d, 1H), 6.66 (dd, 1H), 6.52 (d, 1H), 6.07 (s, 1H), 5.53 (t, 1H), 4.67(d, 2H), 3.42 (q, 4H), 1.11 (t, 6H). ^13^C NMR (100 MHz, DMSO-*d*_6_, ppm): *δ* 161.20, 156.93, 155.65, 150.18, 125.09, 108.53, 105.68, 103.87, 96.76, 59.07, 43.99, 12.35.UV-vis: *λ*_max_ = 375 nm; Anal.: calcd for C_14_H_17_NO_3_ (247): C, 68.02; H, 6.88; N, 5.67%. Found: C, 64.67; H, 5.53; N, 5.18%.

### 2.3. Synthesis of (7-Diethylaminocoumarin-4-Yl)Methyl Succinate (DEACMS)

First, 7-diethylamino-4-hydroxymethylcoumarin (1) (2.47 g, 10 mmol), succinic anhydride (2 g, 20 mmol) and DMAP (0.61 g, 5 mmol) were dissolved in CHCl_3_ (100 mL). The reaction mixture was refluxed at 55 °C under vigorous stirring for 24 h. After evaporation of CHCl_3_ under reduced pressure, the residual mixture was washed three times with 1 M HCl and then extracted with saturated NaHCO_3_ solution. The basic aqueous phase was washed with ether and acidified to pH 5.0 with 1 M HCl. The precipitate was collected and dried in vacuum at 40 °C for 24 h to give (7-diethylaminocoumarin-4-yl)methyl succinate (DEACMS) as a yellow solid with a yield of 47% M.P. (°C): 205.1.^1^H NMR (400 MHz, DMSO-d_6_, ppm): δ 12.34 (s, 1H), 7.46 (d, 1H), 6.69 (dd, 1H), 6.54 (d, 1H), 6.01 (s, 1H), 5.30 (s, 2H), 3.43 (q, 4H), 2.68 (t, 2H), 2.54(t, 2H), 1.12 (t, 6H). ^13^C NMR (100 MHz, DMSO-d_6_, ppm): δ 173.55, 171.89, 160.67, 155.79, 150.59, 150.45, 125.49, 108.75, 105.24, 104.98, 96.83, 61.32, 44.04, 28.66, 12.35.UV-vis: λ_max_ = 381 nm; Anal.: calcd. for C_18_H_21_NO_6_ (347):C,62.25; H, 6.05; N, 4.03%. Found: C, 60.09; H, 5.34; N, 3.92%.

### 2.4. Synthesis of Carboxymethyl Chitosan-(7-Diethylaminocoumarin-4-Yl)Methyl Succinate (CMCS-DEACMS)

The carboxymethyl chitosan-(7-diethylaminocoumarin-4-yl)methyl succinate (CMCS-DEACMS) (CMCS-DEACMS) was synthesized in a similar procedure according to the literature [43]. Typically, 20 mL of hydroalcoholic (water/EtOH~1:1, v/v) solution of CMCS (100 mg) was prepared. Meanwhile, DEACMS, EDC and NHS (molar ratio of DEACMS:EDC:NHS~1:1:1) were added into DMSO (5.0 mL) in sequence, and the reaction mixture was stirred at room temperature for 1 h to activate the carboxyl groups of DEACMS. Then, the mixture was dropwise added into the CMCS solution and stirred at room temperature for 24 h. The resultant mixture was extensively dialyzed in deionized water for 48 h. Finally, CMCS-DEACMS were obtained by lyophilization. All the above procedures were performed in the dark. Three samples with varying molar ratios of DEACMS to glucosamine units of CMCS~1:3, 1:5 and 1:7 were prepared and labeled as CMCS-DEACMS-1, CMCS-DEACMS-2 and CMCS-DEACMS-3.

### 2.5. Preparation of CMCS-DEACMS Micelles

The dry CMCS-DEACMS (10.0 mg) was dissolved in 10 mL deionized water, and the solution was stirred at room temperature for 6 h, followed by sonication using a probe-type ultrasonicator (Scientz-IID, Ningbo Xinzhi Bio-tech Co., Ningbo, China) at 350 W using a pulse function (pulse on 2.0 s, pulse off 2.0 s) in ice-bath for 10 min. The obtained micelle solution was filtered through a 0.45 μm Millipore filter to remove larger particles. All operations were carried out in the dark.

### 2.6. Characterization of the CMCS-DEACMS Micelles

The critical micelle concentration (CMC) value of the amphiphilic CMCS-DEACMS was determined by fluorescence measurement using pyrene as a probe [44]. Pyrene was firstly dissolved in acetone, and 0.5 mL of 6 × 10^−6^ M pyrene solution was added to volumetric flasks. After the acetone was evaporated under 50 °C, 5 mL of fresh CMCS-DEACMS solutions at concentrations ranging from 0.2 to 2000 μg/mL were added into the aforementioned volumetric flasks of pyrene. The solution was ultrasonicated for 30 min and shaken overnight to reach the solubilization equilibrium of pyrene before test. Then, the fluorescence spectra of solutions were recorded on a Cary Eclipse fluorescence spectrophotometer (Agilent, Santa Clara, CA, USA) at room temperature. The excitation wavelength was 333 nm, and the fluorescence emission spectra were recorded from 350 to 600 nm. From the pyrene emission spectra, the intensity ratio of first peak (*I*_1_, 373 nm) to third peak (*I*_3_, 383 nm) was used to estimate the polarity of the pyrene microenvironment [45]. By the profile of *I*_1_/*I*_3_ as a function of concentration (in logarithmic scale), the CMC value was determined from the junction of the horizontal tangent through points at low concentrations with the tangent to the curve at the inflection point.

The hydrodynamic diameter of the CMCS-DEACMS micelles was determined by dynamic light scattering (DLS) experiments with a laser light scattering spectrometer (BI-200SM, Brookhaven, MS, USA). The measurements were performed with the scattering angle of 90° at 25 °C. All the solutions were filtered through a 0.45 μm Millipore filter before DLS experiments. Morphological evaluation was performed on a JEM-2100 transmission electron microscope (JEOL, Tokyo, Japan). The NMR measurements were carried out on a JNM-ECZ400 spectroscope (JEOL, Tokyo, Japan). The FTIR spectroscopy was performed on a Thermo Scientific Nicolet iS50 spectrometer (Thermo Fisher, Madison, WI, USA), and the spectra were recorded using 32 scans over a 4000–650 cm^−1^ range. The decomposition pattern of samples was analyzed by using a TG209F1 Libra Thermal Gravimetric Analyser (Netzsch, Selb, Germany). The samples were heated from room temperature to 800 °C at a rate of 10 °C/min using nitrogen gas (N_2_). The UV-vis absorption and emission spectra of CMCS-DEACMS solution were recorded on a UV-2700 UV/vis spectrophotometer (Shimadzu, Kyoto, Japan) and an Agilent Cary Eclipse spectrofluorometer (Agilent, Santa Clara, CA, USA), respectively. The Stokes’ shift was calculated from the difference in the absorption and emission maxima of the micelles [28].

### 2.7. Photoresponse and Stability of the CMCS-DEACMS Micelles

A metal halide lamp (MH400, Bashida Lighting Electric Co., Lanxi, China) was used as simulator to mimic natural sunlight, and the distance between the lamp and micelles was set as 40 cm. The photo response of the CMCS-DEACMS micelles was monitored by UV-vis, fluorescence, DLS and TEM measurements after irradiation under simulated sunlight. The micelle solutions kept in the dark were used as a control.

### 2.8. Preparation of 2,4-D Loaded Micelles

The 2,4-D loaded CMCS-DEACMS-2 micelles were prepared according to the literature [22,46]. CMCS-DEACMS (10.0 mg) was dissolved in buffer solution (pH 7.4) to obtain a concentration of 1 mg/mL. Then, 1 mL of 2,4-D/methanol solution (12 mg/mL) was added dropwise into the CMCS-DEACMS-2 micelle solution under vigorous stirring for 1 h, followed by ultrasonic treatment for 10 min. Then, the whole solution was transferred into a dialysis bag (MWCO, 3.5 kDa) and dialyzed against the same buffer for 12 h. Finally, dialyzed products were filtrated through a 0.45 μm Millipore filter and lyophilized to obtain 2,4-D loaded micelles.

The amount of 2,4-D encapsulated in the micelles was determined by HPLC (LC-2030, Shimadzu, Japan) using a C18 bonded reverse phase column with a mobile phase of methanol–water mix (60:40, pH 3.0) at a flow rate of 1.0 mL/min. The peak was detected at 285 nm with a UV detector. The encapsulation efficiency (EE) and loading content (LC) of 2,4-D in the CMCS-DEACMS micelles were calculated as follows [47]:(1)$$\mathrm{EE}\;(\%)\;=\;\frac{\mathrm{weight}\;\mathrm{of}\;\mathrm{loaded}\;2,4\mathrm{-D}}{\mathrm{weight}\;\mathrm{in}\;\mathrm{feed}}\;\times 100\%$$
(2)$$\mathrm{LC}\;(\%)\;=\;\frac{\mathrm{weight}\;\mathrm{of}\;\mathrm{loaded}\;2,4\mathrm{-D}}{\mathrm{weight}\;\mathrm{of}\;\mathrm{micelles}}\;\times 100\%$$

### 2.9. Controlled Release of 2,4-D Loaded Micelles

Lyophilized 2,4-D loaded micelle samples (5.0 mg) were dissolved in 5.0 mL of buffer solution (pH 7.4) and then placed into a dialysis bag. The dialysis bag was immersed in 100 mL of the same buffer. The system was exposed to simulated sunlight under mild shaking at 25 °C. At predetermined time intervals, 1 mL dialysis solution was withdrawn and analyzed by HPLC; then, 1 mL fresh buffer was added back. The samples without irradiation upon simulated sunlight were also used as a control. The cumulative amount of released 2,4-D was calculated against time according to the following equation:(3)$$\mathrm{Cumulative}\;\mathrm{release}\;(\%)\;=\frac{V_{e}\sum_{i=0}^{n-1}C_{i}+\;V_{0}C_{n}}{m_{\mathrm{pesticide}}}\times 100\%$$
where *V*_e_ and *V*_0_ are the sampled volume taken at a predetermined time interval (*V*_e_ = 1 mL) and the volume of release solution (*V*_0_ = 100 mL), respectively; *C*_n_ (mg/mL) is the 2,4-D concentration in the release medium at time *n*. The *m*_pesticide_ (mg) is the total amount of pesticide encapsulated in the micelles.

### 2.10. Bioactivity of 2,4-D Loaded Micelles

The bioactivity of 2,4-D loaded micelles was evaluated by taking advantage of one target dicot plant cucumber (*Cucumis sativus L.*) and one nontarget monocot plant wheat (*Triticum aestivum L.*), according to the laboratory bioassay of 2,4-D nanocarriers reported by Cao et al. [48]. For cucumber bioassay study, similarly sized germinated seeds were placed in a Petri dish with a filter paper moistened with 10 mL of 2,4-D loaded micelle solutions at a concentration of the active ingredient equivalent to the field application rate (0.6 kg/ha). Control plants were also grown in Petri dishes and treated with the same amount of free 2,4-D and deionized water. The Petri dishes were incubated in a room with humidity of around 80%. Day and night temperatures were around 28 and 20 °C, respectively. All the Petri dishes were exposed to natural sunlight, simulating the open field planting. Additionally, the Petri dishes were exposed to simulated sunlight for 2 h everyday throughout the experimental period. Same amount of distilled water was applied to each Petri dish for daily watering. After 10 days, the root length and fresh weight were recorded to assess the bioactivity of the target plant.

The bioactivity for the nontarget plant wheat was tested in pot experiments. Ten seeds of wheat were sown in each pot (17.0 cm high with diameter of 15.0 cm) and grown in the same room as cucumber. Same amount of distilled water was applied to each pot for daily watering. One week after sowing, the 2,4-D loaded micelle solution was applied postemergence at an application rate of 2.5 kg/ha (the maximum concentration recommended for field application of 2,4-D) [49]. The samples treated with same amount of free 2,4-D and deionized water were used as controls. After another week, the plant height and fresh weight of the aerial part of the wheat were measured to monitor the response of nontarget plants to the herbicide nanocarriers.

## 3. Results and Discussion

### 3.1. Synthesis and Characterization of CMCS-DEACMS

The light-responsive carboxymethyl chitosan-(7-diethylaminocoumarin-4-yl) methyl succinate (CMCS-DEACMS) conjugate was synthesized from a three-step route starting from commercially available chemicals, as shown in Scheme 1A. Oxidation of 7-diethylamino-4-methylcoumarin with SeO_2_ and the following reduction by NaBH_4_ afforded the intermediate **1**. Then, 7-Diethylamino-4-hydroxymethylcoumarin (DEACMS) was prepared by the esterification reaction of **1** with succinic anhydride, with DMAP as the catalyst, according to the literature [50,51]. The chemical structures of the products at each step were characterized with NMR (Appendix A from Appendix A). Then, the CMCS-DEACMS was synthesized by the reaction of the carboxyl group of DEACMS with the amine group of CMCS via an amide linkage using EDC/NHS as coupling agents.

Formation of CMCS-DEACMS was characterized by ^1^H NMR and FTIR (Figure 1A). In the FTIR spectrum of CMCS, the 3300 cm^−1^ broad absorption band was attributed to the overlapped stretching vibrations of O–H and N–H. The absorption bands at 2890 cm^−1^ and 1322 cm^−1^ could be assigned to the respective C–H stretching and bending vibration [43]. The absorption peak at 1051 cm^−1^ represents the stretching vibration of C-O, indicating that the carboxymethyl group mainly occurred in the C-6 position [52]. Two absorption bands at 1589 cm^−1^ and 1416 cm^−1^ could be assigned to the asymmetric and symmetric stretching vibrations of –COO^−^, respectively [53]. Compared with CMCS, a new shoulder peak assigned to the carbonyl of ester group was observed at 1698 cm^−1^ in the spectrum of CMCS-DEACMS. The appearance of an absorption band at 2929 cm^−1^ corresponded to C–H stretching of DEACMS.

The chemical structures of CMCS and CMCS-DEACMS were further confirmed by ^1^H NMR, as shown in Figure 1B. The proton signals of CMCS in D_2_O were in accordance with previous reports [22]. The resonances at 3.1–3.8 ppm were assigned to the ring protons of the CMCS backbone (H3, H4, H5, H6), and the peaks at 2.3–2.5 ppm were ascribed to the proton of H2 [43,54]. Chemical shift values at 1.88 ppm and at 4.3 ppm corresponded to the protons of the acetamido group and –CH_2_COO^−^ substituted on the C6 hydroxyl group [54,55]. Compared to CMCS, the appearance of new signals in the region of 6.0–7.5 ppm for CMCS-DEACMS corresponding to the phenyl ring of DEACMS confirmed the successful chemical linkage of DEACMS to CMCS. All these results confirm the successful conjugation between DEACMS and CMCS.

The degree of substitution (DS) can be defined as the number of DEACMS per 100 glucosamine units of CMCS. The DS of DEACMS was determined by the peak areas of the phenyl ring of DEACMS at *δ* 6.00 to 8.00 ppm, acetamido group of CMCS (*δ* 1.88 ppm) and the deacetylation degree of CMCS [53,56]. The DS of DEACMS on CMCS were 10.6%, 7.3% and 3.5% for DEACMS-CMCS-1, DEACMS-CMCS-2 and DEACMS-CMCS-3, respectively. Apparently, it increases with the increase in the feed ratio of DEACMS and CMCS.

Figure 2 displays the TGA thermograms of CMCS and CMCS-DEACMS. It can be seen that both CMCS and CMCS-DEACMS mainly underwent two decomposition stages. The main loss in the first stage up to 150 °C was due to the loss of adsorbed water. The second decomposition of CMCS, starting from 200 to 800 °C, was due to the decomposition of the polysaccharide [39]. The DTG curves show that the maximum decomposition rate of CMCS occurred at the peak of 267 °C, whereas the CMCS-DEACMS displayed a maximum weight loss rate at 296 °C, which indicates that the thermal stability was enhanced after introduction of the pendant DEACMS groups.

### 3.2. Self-Assembly Behavior of CMCS-DEACMS Micelles

The synthesized CMCS-DEACMS conjugate is a typical amphiphilic polymer containing both hydrophobic coumarin moieties and a hydrophilic CMCS backbone. When the concentration is above CMC, it will self-assemble into a core-shell micelle structure in aqueous media, as illustrated in Scheme 1B. The CMC of the CMCS-DEACMS conjugate was investigated by using pyrene as a fluorescence probe. Pyrene molecules preferably locate in the hydrophobic region of CMCS-DEACMS micelles rather than the aqueous phase due to their strong hydrophobic nature and very low solubility in water [33]. The variation in the intensity ratio of *I*_1_ (373 nm) to *I*_3_ (383 nm) is quite sensitive to the microenvironment where pyrene is located [45]. Figure 3A shows the profile of *I*_1_/*I*_3_ against polymer concentration for CMCS-DEACMS-2, and the abrupt decrease in *I*_1_/*I*_3_ corresponds to CMC. The CMC values were determined by the crossover point to be 0.013, 0.027 and 0.042 mg/mL for CMCS-DEACMS-1 (Appendix A), CMCS-DEACMS-2 and CMCS-DEACMS-3 (Appendix A), respectively. The CMC values are comparable with other CMCS-based amphiphilic copolymers [22,44]. The decrease in CMC with the increase in the DS of DEACMS can be attributed to the higher content of the hydrophobic coumarin component in the CMCS-DEACMS, which resulted in fewer molecules being available to form stable micelles. This phenomenon is consistent with previous results reported by Yan et al. [23].

The morphology and hydrodynamic diameter of the self-assembled CMCS-DEACMS-2 micelles were characterized by TEM (Figure 3B) and DLS (Figure 3C), respectively. According to TEM images, the CMCS-DEACMS-2 formed spherical micelles with diameters of around 70 nm. DLS results showed a monomodal size distribution with average hydrodynamic diameters of 109 nm, a larger value than the size obtained from the TEM experiment, which was due to shrinkage of the micelles during the drying process of the TEM samples. The samples were further monitored for up to 10 days, showing a slightly higher value of around 113 nm, suggesting temporal stability.

### 3.3. Light-Responsive Property of the Micelles

The CMCS-DEACMS micelles in aqueous solution were exposed to simulated sunlight at regular intervals, and the photochemical reaction was followed by UV-vis and fluorescence spectroscopy (Figure 4). The absorption and emission spectra of DEACMS are shown in Appendix A as a contrast. In the UV-vis spectra of CMCS-DEACMS, one prominent absorption band with a maximum at 391 nm can be observed, a spectral feature corresponding to the absorption of the coumarin units. These results are in accordance with other polymers with photocleavable coumarin units [14,30]. As expected, the CMCS-DEACMS sample is photochemically active in the visible region due to the electron-donating diethylamino groups [29,57]. Under irradiation with simulated sunlight, the intensity of peak absorption decreased and maximum absorption wavelength hypsochromically shifted to 378 nm, indicating that the photolysis reaction took place. Further, the emission spectra showed that the CMCS-DEACMS micelles are fluorescent in nature, with a maximum emission wavelength at 500 nm and a Stokes’ shift of 109 nm. Similar emission intensity decreases and hypsochromic shifts from 500 nm to 488 nm were also detected in the fluorescence spectra. The increasing trend in the intensity of emission maxima upon light irradiation is in accordance with previously reported polyphosphazenes with pendant coumarin groups [30]. The hypsochromic shift of the absorption and emission maxima indicates the decomposition of the coumarin-containing CMCS-DEACMS to the corresponding photoproducts **1**, whose maximum absorption and emission wavelength were 375 nm and 457 nm (Appendix A), respectively. On the contrary, aqueous solution of CMCS-DEACMS micelles showed negligible change during 24 h in the dark both in the UV-vis and emission spectra, indicating its stability in the dark (Appendix A). Further, we also monitored the photolysis of CMCS-DEACMS micelles using ^1^HNMR spectroscopy. The signals at 6.0–7.5 ppm ascribed to the coumarin groups can hardly be observed after irradiation under simulated sunlight, indicating the occurrence of the photocleavage reaction (Appendix A).

The mechanism of the photolysis of the CMCS-DEACMS involves initial heterolysis of the C-O ester bond (photo S_N_1) to produce an ion pair of coumarinyl methyl carbocation and a carboxylate anion on *N*-substituted CMCS. Finally, after separation of the ion pair in water, the methylenic carbocation is trapped by water to yield compound **1**, while the carboxylate anion abstracts a proton from water to yield the corresponding CMCS derivative [14,28], as illustrated in Scheme 2.

The light-responsive properties of the micelles were further tracked by TEM and DLS (Figure 4). After exposure under simulated sunlight for 5 h, the average hydrodynamic diameter of micelles increased from 109 nm to 200 nm, arising from the photocleavage of the coumarin moieties at the micelles’ core. This photolysis reaction left hydrophilic terminal carboxyl groups and resulted in swelling of the micelles, as observed for other coumarin-based micelles or nanoparticles [58,59]. TEM images confirm the light-induced morphological change; micelles appeared to be smaller and the core-shell structures were still maintained even after irradiation under simulated sunlight.

### 3.4. Light-Controlled Release of 2,4-D

Pesticide carriers with controllable release in response to environmental stimuli are highly attractive in terms of smart “on-demand” delivery and increased efficacy. In previous work, the utilization of amphiphilic micelles for pesticide loading and controlled release has been reported [22,33,60]. In the present work, CMCS-DEACMS-2 was utilized to investigate the pesticide loading and light-responsive release profile by using herbicide 2,4-D as a model pesticide. The EE and LC of the 2,4-D loaded CMCS-DEACMS micelles were found to be 7.25% and 8.7%, respectively. The EE and LC can be readily regulated through adjusting the concentration of the 2,4-D/methanol feed solution. Other LC values were also determined to be 6.72% and 9.54% for 2,4-D/methanol feed solutions with concentrations of 8 mg/mL and 16 mg/mL, respectively. This indicates that a high concentration of the initial pesticide solution would result in a high LC [33]. The light-responsive release curves are presented in Figure 5. The simulated sunlight used in this work was applied as a commercial plant growth light with a spectral output of 320–800 nm, among which the CMCS-DEACMS micelles were photochemically active. Without irradiation upon simulated sunlight, around 65% of the loaded 2,4-D is released within 500 min. However, with irradiation upon simulated sunlight, approximately 90% of the loaded 2,4-D is released within the same period. It is indicated that the release was promoted by irradiation upon simulated sunlight, which derived from the rapid escape of 2,4-D from the destabilized CMCS-DEACMS micelles due to the photocleavage of coumarin groups.

### 3.5. Bioactivity Activity

The bioactivity of the 2,4-D-containing CMCS-DEACMS micelles was compared to free 2,4-D and deionized water in laboratory using one dicot target plant cucumber (*Cucumis sativus L.*) and one monocot nontarget plant wheat (*Triticum aestivum L.*), according to a previous report [48]. The 2,4-D-containing micelles showed effective root growth inhibition of cucumber plant, as well as free 2,4-D (Figure 6). After 10 days of experimentation, the 2,4-D-containinh CMCS-DEACMS micelles displayed similar root growth inhibition compared to free 2,4-D, and the root length and fresh weight was reduced to around 23% and 50% compared to the control plant with deionized water, demonstrating the good herbicidal bioactivity of the 2,4-D-loaded CMCS-DEACMS micelles.

Wheat was selected as a model monocot plant to evaluate the safety of 2,4-D loaded micelles, since 2,4-D is safe for monocot plants under the recommended dosage. At the maximum 2,4-D dosage recommended for field application (2.5 kg/ha), both the 2,4-D-containing micelles and free 2,4-D applied post-emergency had no harmful effect on the plant height and free weight (Figure 7). Earlier reports have also shown that 2,4-D nanocarriers do not affect the growth of nontarget plants (wheat or *Zea mays*) [40,48]. Therefore, the prepared CMCS-DEACMS micelles can be recognized as desirable herbicide nanocarriers for nontarget monocot plants.

## 4. Conclusions

A novel light-responsive amphiphilic CMCS-DEACMS nanocarrier for pesticide was successfully prepared by introducing hydrophobic and photolabile coumarin groups onto the main chain of hydrophilic CMCS. The chemical structures were characterized by FTIR and NMR. The amphiphilic CMCS-DEACMS can be self-assembled into spherical micelles in aqueous solution through the hydrophobic interaction between the DEACMS groups. The destabilization of the micelles in aqueous solution under simulated sunlight irradiation leads to accelerated 2,4-D release from the micelles. Further, good bioactivity was observed on the target plant with no impact on the nontarget plant. Hence, this new type of light-responsive CMCS-DEACMS micelle as a nanocarrier holds great potential for smart pesticide delivery in the agricultural industry.

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
