# Peer review of "Coumarin-Containing Light-Responsive Carboxymethyl Chitosan Micelles as Nanocarriers for Controlled Release of Pesticide"

_polymers, 2020, doi:10.3390/polym12102268_

Round 1

Reviewer 1 Report

The evaluated manuscript entitled „ Coumarin-containing light-responsive carboxymethyl 2 chitosan micelles as nanocarriers for controlled 3 release of pesticide” constitutes an interesting report on the results of research devoted to amphiphilic biopolymers prepared by introducing hydrophobic (7-diethylaminocoumarin-4-yl)methyl succinate (DEACMS) onto the main chain of hydrophilic carboxymethyl chitosan (CMCS) via formation of an amide bonding, which were able to self-assemble into spherical polymeric micelles in aqueous media, and were utilized as light -responsive nanocarriers for controlled release of pesticide.

All experimental protocols and methods, together with characterization data and supporting spectra are presented in a logical and complete fashion. Relevant prior works are presented approreately. Therefore, the manuscript deserves publication. However , for those interested in this paper, it would be useful to find additional informations mainly related to the synthetic protocols included to the manuscript.

Thus, protocol   2.2. Synthesis of 7-Diethylamino-4-hydroxymethylcoumarin (1)( lines 1017-120) should be supplemented by adding the melting point and detailed chemical shift values ( for 1H and 13 C-NMR spectra) . UV-VIS data should also be added ( this spectrum is shown as Figure S7 in Supporting Information). A complete elemental analysis should also be performed Similarly, protocol 2.3. Synthesis of (7-Diethylaminocoumarin-4-yl)methyl Succinate (DEACMS) )( lines 121-128) should be supplemented by adding information about the nature of the compound (liquid, solid , semicrystallin). If the compound is solid the melting point should be added .  detailed chemical shift values ( for 1H and 13 C-NMR spectra) . UV-VIS data should also be added. Moreover, the 1H-NMR spectra presented in the Supporting Information ( Figures S1-S3, should be supplemented by the original copies showing the values of the chemical shift of the individual signals. This suggestion is due to the fact that in Figures S1-S3 the signals corresponding to the solvent (DMSO-D6) are very sharp lines (this absorption of unchanged protons present in the solvent molecules, due to the structure of (O) S [CHD2] 2, should be the multiplet).

Reviewer 2 Report

Feng et al. report on the utilization of a coumarin functionalized chitosan (CMCS-DEACMS) as a light sensitive nanocarrier for the pesticide 2,4-Dichlorophenoxyacetic acid (2,4-D). The idea is interesting. Similar systems have been rarely investigated despite their significance for agriculture. The authors investigated in detail the structure and properties of the formed empty and loaded micelles. The light-controlled release behavior of CMCS-DEACMS micelles has been also proven experimentally under very realistic conditions. The plant specificity of the prepared formulations is very important outcome of the work. However, there are some points that need further revision in the manuscript (see below).

  1. Experimental section, photoresponse experiments: how close to real life are the simulated light irradiation conditions utilized?
  2. Micelle loading with 2,4-D: The reported loading capacity is the maximum payload attained in these systems? Have other loading degrees been studied?
  3. The authors should discuss on more detail the choice of the loading protocol utilized. Have other loading protocols been examined?
  4. Some data on the colloidal stability of the formulations vs. time should be added.
  5. Fig. 3: better quality TEM images should be provided.
  6. Light response of the micelles: Are there data regarding the extent of potential degradation of CMCS-DEACMS micelles under the specified irradiation conditions? It would be interesting to determine the degree of change in hydrophobicity of the nanosystem and chemical composition of the macromolecular component under different exposure conditions.
  7. Light controlled release of 2,4-D: some detailed discussion regarding the light exposure protocols is needed, e.g. justification of chosen conditions (duration etc?).
  8. Several typos and format errors (concerning letter size etc.) should be corrected.

Round 2

Reviewer 2 Report

I am satisfied by the responses of the authors to my comments and the revisions made. I suggest acceptance of the manuscript.